# Addressing Sustainable Social Change for All: Upcycled-Based Social Creative Businesses for the Transformation of Socio-Technical Regimes

**DOI:** 10.3390/ijerph17072527

**Published:** 2020-04-07

**Authors:** Sara Calvo, Andrés Morales, Pedro Núñez-Cacho Utrilla, José Manuel Guaita Martínez

**Affiliations:** 1Business and Communication Department, Universidad Internacional de la Rioja, 26009 La Rioja, Spain; andres.morales@unir.net; 2Polytechnic School of Linares, University of Jaén, 23071 Jaén, Spain; pnunez@ujaen.es; 3Business Deparment, Valencian International University, 46002 Valencia, Spain; guaitajo@gmail.com

**Keywords:** upcycling, socio-technical regimes, niches, social and creative enterprises, grassroots organizations, video research

## Abstract

The global challenges caused by socio-economic inequalities, climate change and environmental damage caused to ecosystems, require changes in human behavior at all organizational levels, including companies, governments, communities, and individuals. In this context, it is important to analyse how social and creative companies that work in the fashion and industrial design recycling sector can address sustainable social change. In this paper, we propose an analysis in the countries of the global South. To learn how grassroots innovations can contribute to the development of sustainable strategies, we perform the framework of Technical transitions. We analyze the three main areas of activity that constitute an effective niche construction: social networks, expectations and visions, and learning. A qualitative methodology is used, a video case study with six grassroots organizations in South Africa, Sri Lanka, Malaysia, and Brazil. The results reflect the important role played by these grassroots innovations, contributing to the development of social and creative recycling companies that address socio-economic and environmental problems.

## 1. Introduction 

The global challenges of socio-economic inequalities, climate change, and damage to ecosystems require changes in the behaviour of companies, governments, communities, and individuals [1]. The linear economy, based on the manufacture, consumption, and disposal of basic products following the ‘take-make-dispose’ plan has become a public concern in recent years. This is due to resource-intensive production processes and the increase in the amount of waste sent to landfill [2]. In recent years, there has been a tendency to emphasise the circular economy that follows the 3R approach: reduce, reuse, and recycle. This approach to waste reduction is a priority for international institutions. The 2030 Agenda, the roadmap for sustainable development approved by the international community, includes in Sustainable Development Goal (SDG) number 12, which highlights the importance of sustainable consumption and production modalities [3].

Therefore, the objective of this work is to propose the means to reduce the generation of waste, such as prevention, reduction, recycling, and reuse activities important for the study. More and more waste is generated worldwide. It is estimated that in 2016, more than 44 million tons of waste were generated, which would amount to almost 4500 Eiffel Towers, according to the report of the United Nations University, ‘The Global E-Waste Monitor: Quantities, Flows and Resources’. From this figure, it is estimated that only 20% of the waste generated is recorded as recovered and recycled worldwide. The mentioned report also shows data by region and it should be noted that Asia was the region generating the highest volume of waste at 18%, followed by Europe at 12% of waste generated.

One way to address this problem of waste generation is the application of ‘creative reuse’ or what is known as ‘upcycling’, which is the use of products, waste materials, or wastes to manufacture new materials or products of better quality. It differs from other recycling processes in that this process tends to degrade or decompose the material to be recycled. The ideal of this concept is to give a second chance to objects destined for garbage in a different way, with the aim of obtaining a better result than the original [4,5]. In recent years, we can see enormous attention being paid to recycling and upcycling sustainable related activities. Few studies have focused on the inclusion of energy criteria to measure efficiency in recycling processes in the industrial sector but lacking long-term sustainability approaches, while other studies have highlighted the importance of integrating social, economic, and ecological considerations into the designs of sustainability assessment processes [6,7]. However, there is a critical knowledge gap with regards to the contribution of upcycling-based social and creative enterprises to sustainable social change and, in particular, the role of those organizations that are working within the fashion and industrial design fields [8,9]. 

Social and creative companies are organizations that trade for social and environmental purposes, and they can be included within the creative sector. Therefore, this paper aims to address this research gap by exploring Upcycling-based Social and Creative Enterprises (UBSCEs) that are working within the fashion and industrial design fields in Africa, Asia, and Latin America, trying to develop a good understanding of how these organizations can contribute to sustainable social change. Despite its importance, it is a subject that has not been widely analyzed by the literature [10], and so its analysis is justified and will make it possible to infer information about the behaviour of these companies of a social and creative nature. For this, the authors rely on the socio-technical transitions framework [11], and the three areas of activity that constitute effectiveness in construction niches: social networks, expectations and visions, and learning. The authors explore this phenomenon using the qualitative methodology of case study through video, analyzing six grassroots organizations: three in Africa (Hands of Honour, H18, and Sealand), two in Asia (Biji-Biji and Ecomaximus Initiative), and one in Latin America (La da Favelinha).

This paper raises a series of research questions:

**RQ_1_**_:_ Can we explain the contribution of UBSCEs to the development of a niche with the socio-technical framework transitions?

**RQ_2:_** To what extent do UBSCEs’ experiences and their interaction with networks and intermediaries suggest that niche management is emerging, and at what stage of development is it?

**RQ_3_**_:_ What are the challenges to promote and support the development of a niche within the current system regime?

Regarding the structure of this document, following this introduction, the theoretical framework of the research is presented. The next section presents a discussion about UBSCEs and the differences between the regime and the niche. Subsequently, the methodology designed to illustrate the conceptual framework is exposed. Section 4 illustrates the findings obtained from this research. The final section presents the conclusions of this document and makes suggestions for future research.

## 2. Framework of Socio-Technical Transitions 

### 2.1. A Multi-Level Perspective: Understanding the Micro and Macro Structures

The transitions between socio-technical systems pose the evolution from one model to another. In the current context, both technologies and artefacts play an important role in social functions. However, by themselves, technologies and artefacts have no power, and they do nothing. Only an association with human agency, social structures, and organizations make the devices work [12].

For the study of these aspects, it is appropriate to use the double unit of analysis—technical and social. At the level of social functions, a group of elements work together to achieve full functionality, such as technology, regulation, users, cultural meanings, infrastructure, network maintenance, and production systems. This cluster of elements is named the socio-technical system [12].

Due to the potential for analysis, there is a growing interest in the governance of socio-technical transitions, to learn how modern industrial societies can try to move towards more sustainable development [13]. In recent years, literature on sustainability transitions in the specific field of business organization and management has emerged that poses a multi-level perspective (MLP) to capture the dialectical relationship between micro-level actors and macro-level structures (landscapes, regimes, and niches) [14].

The socio-technical regime means a set of relatively stable and aligned rules that direct the behaviour of a group of actors along the path of incremental innovation. On the other hand, niches are conceived as protected spaces where new socio-technical configurations are established (often as a direct response to an unsustainable regime), experimented with, and developed, away from the normal selection pressures of the regime [15]. Moreover, a landscape can be defined as exogenous events and trends that shape niche-regime dynamics. 

Seyfang and Smith [16] distinguished two types of niche innovations: market based and grassroots. They argued that market-based innovations differ from grassroots innovations in terms of context (market vs social economy), driving force (profit vs social need), niche (market rules vs values), organizational form (firm vs diversity of organizations), and resources (commercial oriented vs diverse non-profit funding). For this paper, the authors are interested in exploring grassroots niche innovations.

The multi-level perspective of socio-technical transitions and innovation [14] addresses the transformations of the basic characteristics of production and consumption systems, for example, the agri-food system, the transport system, the energy system, the financial system, the housing system, and so on. This perspective comes from a combination of disciplines that include history, evolutionary economics, institutional theory and studies of science, technology, and society. In addition, from this approach, apart from the dominant regime in a given system, there are a number of "niches" that are spaces in which social practices and alternative configurations occur. These are spaces where new ideas, models, and ways of doing arise that can influence, be part of or even replace a certain regime at a given time. They are spaces of experimentation, which have different characteristics from those of the regimes, in different dimensions: they operate under other guiding principles and/or privilege other technologies, other types of relationships between actors, channels, forms of knowledge, and so on.

The regimes try to remain stable, although they are under pressure of long-term trends, called the “landscape”, pressures caused by environmental changes (climate change), demographic trends (urbanization), social and political ideologies (neoliberalism), established social values (consumerism), and macroeconomic patterns (globalization). According to Geels [11], transitions occur when the niches arrive at the right time through learning, process improvement, and the support of powerful groups. In addition, the regime is sufficiently pressed by the landscape, and the destabilization of the regime creates windows of opportunity for niches to produce changes in the regime.

Geels [11] characterizes these interactions between regimes and the niche, noting that they depend on several factors, such as the type of landscape, the latter’s pressures on the regime, and the maturity of the niche. For example, when landscape pressure is moderate, but the niches are not mature enough to change the regime, it changes some of its characteristics, but without substantial changes in the system.

When there is much pressure on the landscape and a niche is very mature, it can replace the regime in a short time. According to the strategy style of the niches, Geels [11] identifies several attitudes: firstly the reformist attitude, which seeks to encourage the elites to accept gradual changes “from above”, to change the regime little by little; secondly, the impatient revolutionary, who seeks a change in the elites so that the new expert elites promote drastic changes in the regime; thirdly, the base fighter, who seeks to generate structures parallel to the system, with the hope that they will extend through the example and gradually replace the regime; and fourthly, the revolutionary patient, who seeks to prepare innovations and alternative practices in niches, waiting for the collapse of the system making these practices able to replace those of the regime quickly. In this paper, the authors consider that the UBSCEs are part of a niche that hopes to gradually replace the regime [17].

### 2.2. Upcycled-Based Social and Creative Enterprises: Understanding the Niche and the Regime

In this paper, the authors consider those companies that work in the creation of products within the industrial and fashion sector—the regime. These companies are part of the linear economy, where the manufacturing, consumption, and disposal of basic products are the values of production. Upcycling related initiatives and companies, however, have been shown as a practice within an alternative niche to a regime, that of the industrial and fashion sector. 

Creativity and creative industries are increasingly used in academic literature [18,19]. In a constantly changing industry, such as the industrial and fashion sector, design needs to continuously innovate, offering new products and trends. Innovation involves the process of developing and applying a new idea [20]. According to Varis and Littunen [21], this is the elixir of life for companies, regardless of their size or other attributes of the organization, since the growth, success, and survival of the company depends on its capacity for continuous innovation over time [22]. This is especially significant in the industrial and fashion sector, which varies season to season and requires creativity as a critical ability to compete in this environment. This sector, with its seasonal cyclical demand, requires intense creativity in a short cycle of time that is repeated every three months [23].

The dimensions of this sociotechnical regime can be characterised as follows: The guiding principle of this model would be to offer the consumer a wide variety of producers, optimizing production and cost benefit. The system uses technology intensively and natural resources. The industrial structure of the regime has to do with intensive production, specialized globally, sustained in large global commercial operations. This structure shows increasing levels of concentration of power in a few large groups of companies and technology providers. The access channel to products is increasingly based on purchases in large areas, and they have distribution channels controlled by a few large companies. In the field of policies and regulations that support the regime, the large dominant groups seek deregulation of global markets, a regulation more favorable to mass production.

Upcycling initiatives and companies are a practice that can be framed in a niche that proposes networks to produce alternative products. These initiatives are committed to the use of and respect for existing resources and the environment. This, in recent years, is being seen as a space of maximum interest for the transformation of the economy towards a more just and responsible model [24]. The initiatives in this niche would be prefiguring a new development model, based on more democratic societies and more responsible citizenship [25,26].

That is why members of this niche need to create alternative channels that take into account the reduction of garbage, the use of what already exists, and sustainability. Table 1 shows the characteristics of the socio-technical practices of the regime and the niche. 

There are few studies that have explored UBSCEs. A study conducted in the United Kingdom by Singh et al. [7], in 2019, highlights that, although upcycling increases the quality and lifetimes of materials and products, reduces waste, creates employment opportunities, and encourages sustainable consumer behavior, this activity is largely considered a niche practice. One of the important gaps in the current state of knowledge on upcycling is a lack of systemic understanding about challenges and success factors relating to scaling up upcycling businesses. The results identified the importance of collaboration across the upcycling value chain, involving a wide range of actors. Another study by Jayasinghe et al., in 2019, in Sri Lanka [9] suggested that improved materials provision, communication and education to raise public awareness and product acceptance, quality control of products, creating new markets, and financial support for upcycling are priority interventions for scaling up upcycling social and creative social enterprises in Sri Lanka. 

### 2.3. Niche Construction Activities: Social Networks, Expectations, and Learning

The socio-technical transitions framework has been developed to further understand and govern the processes of niche creation [27]. It allows us to test the role of grassroots innovations (which in our case are those implemented within recycled activities) as agents of change and their ability to form niche spaces where new ideas and practices can be developed [14,28]. On the other hand, Schot and Geels [27] identified three areas of activity that constitute an effective niche construction by observing the conditions under which the niches become influential: social networks, expectations, and visions and learning. First, a growing social network, including all relevant types of actors within the niche, creates opportunities for stakeholders to interact in a micro market that provides temporary protection and the resources necessary for experimentation. Secondly, the articulation of expectations and visions of high quality by the participating actors—particularly when they are robust and shared by many actors—give direction and legitimacy to the niche [28]. Thirdly, and most importantly, there are learning processes in multiple dimensions. Learning processes should not be limited to first-order learning, i.e., data collection, but should also include second-order learning, which is aimed at changing cognitive frameworks and assumptions [27]. The socio-technical transitions framework argues that the better these three processes and their interactions are handled, the greater the chances that the niche will become a market niche, transforming or becoming a viable alternative to an existing regime [28].

As can be seen in Figure 1, there are four main phases in the development of shared technological knowledge: first, a local phase (a set of isolated projects); second an interlocal phase (a niche level emerges where projects share knowledge and experiences); third a trans-local phase (where actors play a role in the development of interest to manage external expectations and local knowledge is systematically fed to constitute the required aggregate learning at the niche level), and a global phase (where we see a greater institutionalization and standardization of practices in the field with niche standards that shape local practices, becoming a stable regime). 

As with other transition theories, this framework has not been without criticism. Several authors have commented that the theory is vague, since there are problems to explain the development of the niche, and since there is little evidence that the initiatives become important learning vehicles for a broader shift towards new socio-technological regimes [29,30]. However, the authors believe that this paper can contribute to the existing literature by generating useful ideas about the niche development stage, and the challenges facing the social and creative business niche based on upcycling within the current system regime, using an approach through a multiple video case study [30].

## 3. Methodology

We decided to study the case of design in the industrial and fashion sector, because it is knowledge intensive and requires the development of creativity to continue. Design is a very important part of sustainability processes [5]. The market thus forces companies to create new materials and products in rapid cycles [5]. Companies tend to address this level of customer pressure by developing design and management processes that support creativity [31]. Therefore, it is a good field in which to study these phenomena.

### 3.1. Data Collection

The findings presented here come from a qualitative investigation of multiple case studies. According to Yin [32], a case study is *“an empirical investigation that analyzes a contemporary phenomenon in depth and within its real-life context, especially when the boundaries between the phenomenon and the context are not clearly evident”* (page 18). The authors use the investigation of case studies on video, since it allows both researchers and interviewees to express themselves creatively, narrating aspects of their experiences or emotions that would not otherwise be told [33]. In addition to this, video research allows us to present the findings in an unconventional way, reaching wider audiences, training participants, and making knowledge accessible and easier to share [34]. To carry out the investigation, three different stages of that collected by video case methodology were approached: first, the preproduction; second, the production; and third, the postproduction [35].

In line with our research design, we applied purposive sampling for case selection suitable for research informed by an existing body of social theory. The study was conducted in three phases over nine months (January to September 2019). In the first phase, a bibliographic search was undertaken to better understand the different dynamics of UBSCEs in the countries selected for the study (Brazil, South Africa, Malaysia and Sri Lanka) (see Table 2). 

Then, the authors identified and selected six organizations as examples of case studies and investigated the activities and development trajectory of each (see Table 3 for more details). The case studies were chosen as they are all included within the social and creative sector and are all working in fashion or industrial upcycling related activities. To capture different environmental contexts, we sampled cases from different countries in Latin America, Africa, and Asia. The authors had previous contact with these organizations, and invited the CEO of each organization to participate in the research study. 

In the second phase, the authors visited each organization and conducted in-depth semi-structured video interviews with key stakeholders (see Table 4 for more details). Each face-to-face semi-structured interview lasted between 30 and 90 minutes, and was video-recorded and transcribed verbatim.

The interview schedule consisted of two sections: the first included questions on the individual characteristics of the participants (for example, gender, age, occupation). The second consisted of open questions related to the characteristics of the organization and in particular, to the three processes indicated in Table 5. In the third phase, field notes were made from observations made of events and organized activities in each of the grassroots organizations, to complement the data collected.

### 3.2. Data Analysis

For data analysis, the authors used a similar procedure as that described by Naber et al [40]. Firstly, an open coding strategy was used to identify the development of a niche. The concepts were used as sensitizing concepts to direct the analysis, meaning that raw data were compared with the theoretical background discussed in the theory section, in particular, the three main processes highlighting the current developmental stage of the niche: social networks, expectations, and visions and learning processes. Key codes were identified from the video interviews, fieldnotes, and secondary sources and were refined as the analysis evolved. An iterative analytical process was used to draw out the key codes, commonalities, and variations across stakeholder responses, and analyzed according to theoretically informed codes using NVIVO. Ethical approval was obtained from participants to publish a video and other research related materials, which included informed consent and “responsible” research practice. A short video was produced using final Cut Studio 3, a professional video and audio program.

## 4. Research Findings and Discussion

The authors applied the theory of socio-technical transitions to first identify the niche development phase that the sector seems to be exhibiting when observing the niche construction processes of social network construction, articulation of visions, and expectations and learning processes. Second, the authors explored the challenges that the niche currently has within the current system’s regime. Table 6 provides a brief overview of how the interview segments were coded, according to our interpretation of the qualitative data. All this took into account the three processes of the socio-technical transitions framework to evaluate the development of a niche: the creation of social networks, the articulation of visions and expectations, and the learning processes.

### 4.1. Theme 1: Social Networking

The growth niches depend on network expansion and networking activities, both internally (creating a sense of community to encourage information exchange) and externally (to attract resources and influence). The specific findings showed that the construction of a social network was particularly important for growth, a basic issue for organizations [41]. The recorded data evidenced that these grassroots organizations participate in network activities in various ways as the availability of a set of national and international partners to share information and experiences. They collaborate with them for the development of projects. When asked to name important partners, respondents will appoint local, national, and international partners, including government departments, foundations, private companies, and universities (see examples of interview transcripts in Table 5). Some respondents commented on their work with actors who have resources, and also commented on how to try to change their agendas and visions. They indicated their ability to attract funds from different institutions, the main one being:


*“We are able to attract funding as it is the momentum. People are starting to get worried about the lack of resources and the damage to our environment”*
Interviewee 8

Evidence was also obtained of networking with organizations internationally. All the case study examples were reported to have social networks at an international level. An example of this strong relationship with international partners is indicated by one respondent who said he worked with international companies to reduce waste and create upcycled products:


*“We have been funded by the British Council to start this upcycling business. This has been a fantastic opportunity to help the community and create awareness about waste”*
Interviewee 3

It was also possible to identify considerable internal networks within the niche, both formal and informal. Respondents mentioned that much of the effort to establish contacts between members have been made through regular meetings and conversations with the leaders of the different projects, to ensure that the work was carried out in a similar line (see Table 5 for details of the transcription segments). Therefore, collaborations are crucial for all the UBSCEs selected for this study. This is in line with a study conducted by Singh et al. [7] that pointed out the importance of collaboration between the different value chain actors of these upcycling initiatives. The authors discovered that social networks are a vital aspect of the development of these grassroots organizations, and that although there is good evidence of contributions to networks worldwide, there is more activity and dependence on pre-existing networks within the project and at the national level [23]. This indicates that the sector is currently in the “trans-local” phase of niche development, moving towards the global level. It is still at an early stage to transform the current system’s regime [11].

However, although the upcycled arts have been recognized by prestigious policy makers and institutions worldwide, several respondents expressed the view that more dissemination was needed to improve the perception of this niche nationally and internationally:


*“We have a lot of experience, but we lack dissemination. We need more support from the government, they are not really interested in the way we are contributing to the environment as well as our contribution providing employment opportunities for people who are coming from disadvantaged background, such as refugees”*
Interviewee 5

Another important finding of our qualitative research was the knowledge of how some of these grassroots organizations have been involved in the creation of new projects, inspiring others and spreading a shared vision (see Table 5 for examples of interview transcripts). The evidence indicates that the objective of the respondents was to achieve wider social changes. It was also observed that there is an influential niche capable of shaping the development of future projects within its shared overview. These organizations, as illustrated in Figure 1, exhibit an area of performance characteristic of the trans-local phase, in terms of shared visions and project expectations, where local knowledge has been fed to constitute the aggregate learning required at the level of niche [11].

### 4.2. Theme 2: Articulation of Visions and Expectations

As the development of shared expectations and visions is considered crucial for a robust niche development, the authors looked for evidence of such visions within the cases analyzed. The findings show that the majority of respondents had a common understanding of the term “upcycled arts” that referred to it as turning waste into something artistically beautiful, as well as creating social impact. However, there were some organizations that were not aware of the ‘upcycling’ concept, and that they were already applying this model within their organizations. As the founder of an organization highlighted:


*“To be honest, we have been working with this idea for a while without knowing that this is upcycling. For what I see, they call it upcycling now, well… we have been doing this for years but we have not used that term. It is good though to know we can use the term to refer to what we are doing”*
Interviewee 11

Respondents confirmed that their work focused on providing employment opportunities for disadvantaged communities, as well as support the environment (see Table 5). The findings also indicate that the participants had very clear visions for their goals and well-defined objectives, since they had maintained the original vision of the organizations and maintained it over time. Respondents reported enthusiastically on the plans they have to continue promoting upcycled artistic activities to keep their legacy. As one of the respondents pointed out:


*“We want to promote upcycled arts to ensure that people move towards the sector. For us is not a product but a social movement”*
Interviewee 5


*“Our vision is to inspire people to get involved in these activities or from their houses, to see that it is possible to create beautiful products caring about the environment, we need to transmit the values of being responsible for the planet, our upcoming generations need to know that”*
Interviewee 7

Table 5 presents examples of interview transcript segments of how they have inspired others propagating a shared vision. Interestingly, another aspect that emerged from the interviews was that these grassroots organizations have been extremely effective in attracting members who not only have the professional skills, but also a shared vision, required to develop their projects:


*“We have a great team, and this has been crucial for the success of our organization. The team is campaigning for the development of a better world. We have been able to develop things through our collective knowledge and similar interest and visions. We are transmitting our employees that it is not only about their salary but the work they are doing!”*
Interviewee 5


*“We have a great team of people from different disciplines but with similar values, well… the environment is great. It is a pleasure to work here; people value the creativity of everyone and support each other, they believe in the importance of being responsible with our planet!”*
Interviewee 9

### 4.3. Theme 3: Learning Processes

Sharing learning is an important activity for this research, since it indicates that the types of learning and the people with whom it is shared vary over time, according to the development phase of the niche. These grassroots organizations showed evidence that learning is shared “upwards” with intermediary organizations that develop projects. The most prominent mechanisms for sharing learning were working with external organizations to produce learning materials and interacting with intermediaries for project development. In addition, most respondents acknowledged that working with organizations that are part of the sector is relevant to their learning experiences (see Table 5 with examples of interview transcripts). Interestingly, part of the learning has not necessarily contributed to the niche, but rather to support another niche, which is an important contribution to the socio-technical transitions framework, suggesting that learning occurs in a single niche regime interaction:


*“We’ve worked even with the local community; an example of this is the work we have done to support upcycling businesses, encouraging local people from slums to develop upcycling business related projects. It has been a great experience, lot of learning!”*
Interviewee 3

A couple of entrepreneurs who had been supported by these organizations corroborated the contribution to other niches (see examples of interview transcripts in Table 5). In addition, respondents accepted that they also actively participated in formal evaluation or monitoring processes, through which learning was consolidated and passed to intermediaries. Another mechanism of reported learning exchange was the exchange of information (i.e., informal contact, ad hoc by phone, emails, or at events) to acquire information and advice. It was reported that one of the learning processes undertaken by other people and organizations was mentoring. The authors also discovered that learning plays an important role within the groups in the development, improvement, and evolution of these initiatives [27]. The most frequent means through which this occurred was “learning by doing”, which took the form of, for example, adapting its activities to better adapt to local contexts and conditions (see Table 5 for more details). 

The evidence indicates that learning has taken place within these case studies, some being “pulled out” by intermediaries, and others “pushed out” by projects themselves through formal evaluations, monitoring, and structured, codified learning mechanisms. In terms of what types of learning is shared, the findings indicate that socio-cultural, human, and organizational aspects were the most common, with the financial aspects being less popular. Therefore, these findings suggest that these organizations are displaying characteristics typical of the third stage of niche development (trans-local phase), where niche-level actors are playing a significant role in the process of aggregating shared learning within the projects themselves. This is very significant to the project’s development and progress, as well as sharing learning with other organizations [11].

### 4.4. The Challenges of Scaling Up: from the Trans-Local to the Global Phase

By proving how useful this theory is to explain developments in grassroots organizations, our analysis reveals that there is indeed evidence of an emerging niche of the type described in the socio-technical transitions framework. In terms of the niche development phases, indicated in Figure 1, they seem to be playing some of the typical roles of the trans-local phase, allowing a wider dissemination by adding project learning and sharing resources with new projects. However, our analysis indicates some possible routes for the development of the niche that passes from the trans-local to the global phase with a series of challenges. Findings from the analysis indicate that there are factors preventing the niche from scaling up from the trans-local to the global phase.

An important point mentioned by all organizations was the competing prices with companies located within the regime. As the founder of a UBSCE commented: 


*“One of our biggest challenges is that we have is to compete with prices with mainstream companies. People don’t understand the amount of work and time dedicated to these products, they care about price, we need to create awareness of our contribution to the environment, it is not an easy task”*
Interviewee 7


*“People often complained that our products are too expensive. I always say to them, this is not only a product but the contribution to the environment you are making, for example reducing plastic in the ocean. We need to support companies that focused on sustainability and care about the future of our planet”*
Interviewee 10

Logically, all organizations are concerned with sustainability issues [4,42]. Additionally, respondents reported difficulties for people when explaining what upcycled arts is outside the organization, and why it is important. This was perceived as a barrier to promote the sector:


*“Not everyone understands the meaning of upcycled arts. Sometimes people are not open to new concepts. We accept these challenges and try to create awareness”*
Interviewee 9

This can be related to a study conducted by Jayasinghe et al. in 2019 [9], which highlighted that one of the main challenges faced by these organizations is the need to raise public awareness. A further point of discussion relates to the scalability of these upcycled related arts grassroots organizations. As one of the respondents stated:


*“We have had difficulties when we are planning in scaling. We are missing the support from the local government”*
Interviewee 3

The research findings highlight that, although grassroots organizations have made great achievements with all the work carried out, and policy makers and international institutions are also showing more interest in the field, important limitations are detected in terms of evolution at the level of the global phase, where the niche could lead to a radical transformation of the regime (see Figure 1). To become a more robust niche, an appropriate management of the experience of the niche is necessary, which influences the perceptions of the regime’s actors, progressively creating cracks in it.

## 5. Conclusions

In this article, the authors have applied the socio-technical transitions framework to test the usefulness of the theory to explain the role of UBSCEs as contributors to the development of niches. This has been examined through the multiple video case methodology to understand to what extent each case shows characteristics of a niche, and to know the potential it has to disseminate and influence the current system [13,28]. The following conclusions can be drawn from this study. First, the document has shown that an emerging niche is evident and is in the trans-local phase [14]. Second, this article provides evidence of the different elements that occur within these organizations: social networks, expectations and visions, and learning processes [27]. The findings indicate that they had very clear (social and environmental) visions and well-defined expectations of their goals and objectives, inspiring other people and organizations to be part of this ‘*social movement’*. In addition, the research findings revealed that sharing learning is important for the organizations studied, as well as for social networks where there is more activity and dependence on pre-existing networks nationwide. However, there is an increase in its network activities internationally.

As a third contribution, this work has shown that there are currently limitations to develop the niche from the trans-local to the global phase. The findings suggest that UBSCEs are currently struggling with their financial sustainability, particularly with the current tensions and price competition between the regime and the niche [14]. This raises questions about whether this emerging sector will develop into a solid niche, and where more support is needed to strengthen the sector to influence the system’s regime. This is particularly relevant within our case study organizations, since the goal is not to completely displace the established regime, but to build a form of parallel infrastructure that aims to provide the necessary systems that individuals cannot provide for themselves. However, we can see that the current demand within the fashion and industrial sector has a more flexible and open approach to the sustainable environment and the importance of reusing materials. 

To conclude, we recognize both the limitations in our research and the scope for future research. While this represents an interesting study to understand the role of UBSCEs in contributing to the development of an upcycled niche, we cannot say that it is generalizable, since it considered a particular phenomenon in a particular location, which makes generalization difficult to apply to other situations. Based on current work, it would be appropriate to look at upcycled arts and grassroots organizations in other Global South countries to examine to what extent they contribute and how they can influence the current regime of the main system. It would also be good to conduct a comparative study between grassroots organizations that work with upcycled arts in different Global North countries and analyze their differences in terms of resources and infrastructure.

## Figures and Tables

**Figure 1 ijerph-17-02527-f001:**
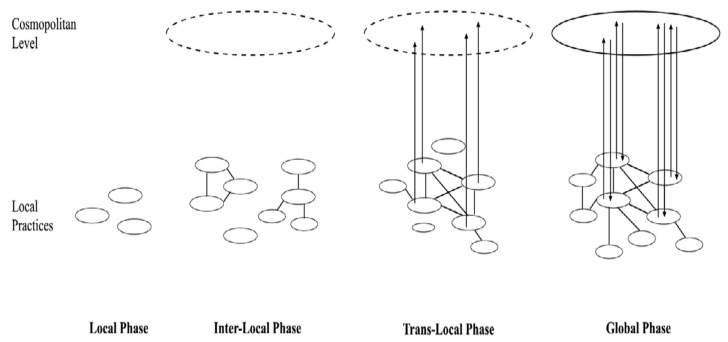
Phase in the development of shared technological knowledge. Source: Based in Geels and Deuten (2006: 269).

**Table 1 ijerph-17-02527-t001:** Comparison between the niche and the regime.

Characteristics	Regime	Niche
Principles of Production	Optimization of production and cost / benefit	Respect for natural limits and balanced relationship with the environment
Technologies	Intensive use of technology	Ecological production
Industrial structure	Intensive production, specialized globally, dependence on producers of fashion, and industrial inputs	Decentralization, diversity of actors, varied production, and linked to the territory
Access channel	Buy in large surfaces of packaged and processed products, distribution channels controlled by a small number of companies	Short marketing channels, minimization of intermediaries, generation of proximity, and trust relationships between producers and consumers
Policies and regulations	Deregulation of global markets, regulations more favorable to the use of biotechnology, support for technology research	Valuation of bio materials, support to organic producers by multiple means, elimination of obstacles for direct marketing
Knowledge sources	Standardized scientific knowledge, produced by the company or research centers with an abundance of resources	Scientific-technical knowledge and peasant and traditional knowledge, empirical, and embedded in the territory.

**Table 2 ijerph-17-02527-t002:** Review of the Literature: Characteristics and context of UBSCEs in the countries selected.

Countries	Characteristics and Context of UBSCEs
South Africa	- There is a growing sector in South Africa. - There has been a recent support from institutions such as the DICE program and Social Enterprise Academy [36].
Sri Lanka	- UBSCEs are growing rapidly across Sri Lanka in sectors ranging from manufacturing to agriculture. - There are some key institutions supporting the development of these initiatives and that includes for example Lanka Social Ventures or Impact Hub. - Although Sri Lanka has no specific policy focus on social enterprises, the national policy framework has, for some time, favored and supported enterprises that generate ‘triple-bottom’ economic, environmental and social benefits [37].
Malaysia	- Upcycling and Creative social entrepreneurship in Malaysia is a growing sector that has the potential to contribute to the socio-economy of the nation. - With the re-establishment of the Ministry of Entrepreneur Development (MED) in July 2018, there has been institutional support available for upcycling social and creative enterprises [38].
Brazil	- The growing role that upcycling social enterprises are playing in supporting disadvantaged communities in Brazil. - There has been support from institutions such as the DICE program and Ashoka [39].

**Table 3 ijerph-17-02527-t003:** Characteristics of grassroots organizations.

Name	Sustainable Impact	Socio-Economic Impact	Link to Website and Video Case Studies
Case 1. Sealand (South Africa).	Fashion Design. A company that produces bags out of plastic collected from the sea.	Employed poor people from Cape Town.	https://www.youtube.com/watch?v=AASEOi3FYJs https://sealandgear.com
Case 2. Hands of Honour (South Africa)	Industrial Design. A company that upcycle obsolete stock and recyclable items.	Employed homeless men.	http://handsofhonour.co.za https://www.youtube.com/watch?v=oLFlJuS3sA8
Case 3. H18 (South Africa)	Fashion Design. A company that focused on the production of cotton crochet products.	Employed disadvantaged women (Refugees).	https://www.youtube.com/watch?v=JqEl07_VaLs&t=129s https://h18.co.za
Case 4. La Da Favelinha (Brazil)	Fashion Design. A company that supports young people with fashion, culture and arts in the slums of Brazil.	Provides employment opportunities as well as creative activities for young people living in slums.	https://www.youtube.com/watch?v=yHD5EDUcSeo
Case 5. Biji-Biji (Malaysia)	Industrial and Fashion Design. A company that champions sustainable living, reuses waste creatively, and loves collaborative production.	Offers volunteering and job opportunities for young people.	https://www.youtube.com/watch?v=H2ziVcW82s4 https://www.biji-biji.com
Case 6. Eco-Maximus (Sri Lanka)	Industrial Design. A company that used elephant dung to enhance the animal’s habitat self-sustainably.	Employed rural women.	https://www.youtube.com/watch?v=u5TbaOPHEns https://www.ecomaximus.com

**Table 4 ijerph-17-02527-t004:** Data collection from semi-structured interviews.

Interviewees	Case	Role	Gender	Interview Duration	Transcript Pages
1	1	Co-Founder	Male	90	11
2	1	Staff	Female	30	4
3	2	Founder	Male	60	8
4	2	Staff	Male	30	5
5	3	Founder	Female	50	7
6	3	Founder	Female	80	9
7	4	Founder	Male	30	5
8	4	Staff	Male	65	9
9	5	Co-Founder	Male	70	10
10	5	Co-Founder	Female	90	12
11	6	Founder	Male	40	7
12	6	Staff	Female	30	4

**Table 5 ijerph-17-02527-t005:** Semi-structured interview script.

Theory	Concept	Interview Questions
General questions		What is your name, gender and age? What is your role at the organisation? How it all started? What has been your role in the organisation? What are the challenges that the organisation is currently facing?
Theory	Social Network Building	Which partners have been involved so far? How was the interaction between partners? Were there enough resources available? Who provided which resources? Did the size of the network increase or shrink?
		Did an overall shared vision emerge?
Theory	Articulation of visions and expectations	How have the expectations evolved? How have expectations been articulated between partners?
		On which experiences were the expectations based?
Theory	Learning processes	What type of learning occurred in the project? How was learning organised? What were the most surprising results?

**Table 6 ijerph-17-02527-t006:** Indicators and values of the theoretical model.

Process	Created Code	Examples of interview (transcript) segments
Social Network Building	National Networks	*“We have links with national government, local government departments, private companies and third sector organizations. All of them support us because of our environmental and social objectives. For example, we receive materials from private companies as they know we are using their waste to create beautiful products”* (Interviewee 7).
		*“We have contracts with private companies that give us their waste to create our products. They do it without expecting anything really”* (Interviewee 10).
	International Networks	*“We have partnerships with international organizations. For example, with our partnership with Minca Ventures Ltd and the British Council, both based from the United Kingdom we are supporting people to develop their own upcycling businesses”* (Interviewee 3).
	Formal and informal internal networks	*“We meet regularly with the team to make sure we are all informed of the progress done and that everybody knows what is happening. We meet regularly as well as share information of companies that are working in the sector and ideas for new products”* (Interviewee 9).
		*“We organize formal and social meetings to keep us updated. There is a good environment in the workplace; people meet to exchange ideas about their projects as well as socialize for lunch. it is like a family, we all care for similar things, so it is great!”* (Interviewee 11).
Articulation of visions and expectations	Shared visions and values	*“We have been recognized by the local community and they trust us, well… they know we are here to create change and make an impact on society and the planet’* (Interviewee 7).
		*“We all have the same vision and values and that is why we came together”* (Interviewee 5).
	Inspiring others and propagating a shared vision	*“We have inspired some people to start up upcycled art related companies. We are very happy about this achievement; we want to inspire other people to reduce waste and being creative and that is our target” * (Interviewee 3).
		*“Some of our staff have identified social and environmental problems in their local areas and have looked for innovative solutions. They have started their own projects to help the local community. it is fantastic to see how our values and vision are affecting other people positively, and how they have become promoters of upcycling initiatives to help their communities”* (Interviewee 7).
Learning processes	Working for other organisations: a key for learning experiences	*“We’ve developed a MOOC of upcycling related arts companies to support start up organizations”* (Interviewee 9).
		*“We’ve worked with other companies developing a wide range of projects. for example, we’ve developed a project with the local government to encourage these initiatives”* (Interviewee 11).
		*“We organize workshops to translate our knowledge to other people so we can inform of our duty to make this world a more sustainable and better place”* (Interviewee 6).
	Contribution to other niches	*“I have personally learned a lot from the staff of other companies that are supporting us and how to manage a business and scale up”* (Interviewee 2).
		*“I have learned a lot; it is difficult to say in words the contribution they have given to my company”* (Interviewee 5).
	Learning by doing	*“We have learned a lot by reflecting on our experiences designing products. for example, we have changed the way we were doing things in here, we are launching new minimum viable products, see if people like them and improve…”* (Interviewee 3).
		*“We give relevance to the knowledge and experience of the local people to design our products, trying to recover their local knowledge. We’ve realized how beautiful this can be if the employees use their local knowledge!”* (Interviewee 11).
		*“We are learning by testing products. When we started our products didn’t have the quality them have now. Now, the products have good quality and feel good to complete in the market”* (Interviewee 12).

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
