# Peer review of "Addressing Sustainable Social Change for All: Upcycled-Based Social Creative Businesses for the Transformation of Socio-Technical Regimes"

_ijerph, 2020, doi:10.3390/ijerph17072527_

Round 1

Reviewer 1 Report

Interesting paper about upcycling. It is good to see how the authors have captured this developing niche and the benefit to the environment, and socioeconomic of the people.

One thing I would love to see is an explanation of the different dynamics among the different countries used in this study. I believe that there are some peculiarities only specific to the different countries and how this has helped to shape the social networking, and interactions of the group of respondents and particularly their workers.

Since the study used a qualitative approach, I will prefer the use of 'findings' instead of 'results' in the analysis and discussion section.

The manuscript should be accepted after attending to the comments made to the authors. 

Author Response

Response to editors and reviewer comments

We would like to thank the editor and reviewers for their constructive comments on our paper and their very useful suggestions for revisions.

We have revised the paper to address fully these outstanding concerns. Specifically we have responded to the key issues raised concerning: (1) an explanation of the different dynamics among the different countries and how this has helped shape the social networking, and interaction of the group of respondents and particularly their workers; (2) the lack of discussion with the previous literature in Section 4; (3) why it is important to understand UBSCEs in fashion and industrial design recycling and (4), an explanation of what kind of case study was performed and how we chose the countries to focus.

Full details of our response to each of the comments made are set out below and all changes made are highlighted in red in the text.

If you require any further information, please do not hesitate to ask.

Yours sincerely

Sara Calvo

Referees comments

Response

Explanation of the different dynamics among the different countries and how this has helped shape the social networking, and interaction of the group of respondents and particularly their workers.

There is limited research about UBSCEs in each of the countries selected. A table has been included with a brief explanation of the different dynamics (see Table 2).

The use of the word ‘findings’ instead of ‘results’.

The term ‘findings’ has been used in the document. The term ‘results’ has been deleted as it is a qualitative study.

Explain why it is important to understand SCEs in fashion and industrial design recycling. What has been done so far and why do we need further development on the topic.

A further explanation why is important to understand UBSCEs has been included in the introductory section as well as in the 2.2 Section ‘Upcycled-based Social and Creative Enterprises: Understanding the niche and the Regime’.

Lack of discussion with the previous literature in Section 4. How do you position within the current discussion and if we find similarities or differences in similar studies conducted in other contexts.

The connection between the existing literature and the findings of this study has been added in Section 2.2.

What kind of case study was performed and how we chose the countries to focus.

The methodology section has been changed significantly. In particular, the authors have explained in more details the type of study conducted and the methods of data collection.

Reflect on the findings: Did you find the same results in all the case studies or there were outlier specific situation due to specific contextual factors? Did you perceive any different perspectives on the topic from different stakeholders belonging to the same case study?

The findings section has been reviewed and changed accordingly. These two questions have been analysed and the differences between different cases and within cases between different stakeholders have been highlighted in the findings.

Reviewer 2 Report

Dear authors,

Thank you for the paper you submitted. The topic is very interesting and the overall manuscript is well papered and structured. Nevertheless, I think the paper could be further improved.

1) Particularly, I have one major issue related to why do we need this study. As the need for addressing upcycling is well explained, I had some difficulties in understanding the focus on social and creative companies in the fashion and industrial design recycling. In this way, I suggest to detail better in the manuscript what has been done so far (Ref [6]) and why do we need further development on the topic. I think that a literature review (also short and narrative) at the beginning of Section 2 could help the reader to better frame the topic.

2) Another issue concerns the lack of discussion with the previous literature in Section 4. I think this issue is somehow related to the previous one. Indeed, as a literature review section on the specific topic is not provided, is not easy for the reader to understand how do you position with respect to previous literature. I am not discussing the soundness of your results, but how do you position within the current discussion and if you find similarities or differences in similar studies conducted in other contexts.

Besides these two main issues, I have some minor comments.

Abstract and 1. Introduction

No minor comment, see major comment 1).

2. Framework

No minor comment, see major comment 1).

I think this Section is very clear and well developed.

3. Methodology

The methodology section is well developed and rich in useful information increasing the reliability of the study.

a) It was not clear to me what kind of case study you have performed (theory generation, theory development, theory refinement). This is however also connected to major comment 1).

b) It was not clear to me how you chose on which Countries to focus

4. Findings and Discussion

Firstly, thank you for the transparency you provided in this section.

See my major comment 2).

Table 5, second row: the quote is not in italic

Table 5, fourth row: I think it is a mistake

I have some questions to ask you that should be intended more as additional food for thought:

a) Did you find the same results in all the case studies or there were outlier specific situation due to specific contextual factors?

b) Did you perceive any different perspectives on the topic from different stakeholders belonging to the same case study? I pass you some papers on this aspect if you are interest

  • Arvai, J., Campbell-Arvai, V., & Steel, P. (2012). Decision-making for sustainability: a systematic review of the body of knowledge. Network for Business Sustainability. https://nbs. net/p/systematic-review-decision-making-for-sustainability- 32e9291b-e3be-41a8-a898-d49cb67752d0
  • Cagno, E., Neri, A., Trianni, A., 2018. Broadening to sustainability the perspective of industrial decision-makers on the energy efficiency measures adoption: some empirical evidence. Energy Effic. https://doi.org/10.1007/s12053-018-9621-0.
  • Cooremans, C. (2012). Energy-efficiency investments an interpretative perspective and energy management. Proceedings of the International Conference on Energy Efficiency in Commercial Buildings. https://doi.org/10.13140 /2.1.4787.5529
  • Eisenhardt, K. M., & Zbaracki, M. J. (1992). Strategic decision making. Strategic Management Journal, 13(S2), 17–37. https://doi.org/10.1002/smj.4250130904
  • Gibson, R. B. (2006). Beyond the pillars: sustainability assessment as a framework for effective integration of social, economic and ecological considerations in significant decision-making. Journal of Environmental Assessment Policy and Management, 8(3), 259–280. https://doi.org/10.1142 /S1464333206002517
  • Thollander, P., & Palm, J. (2012). Efficiency in industrial energy systems: an interdisciplinary perspective on barriers, energy audits, energy management, policies, and programs. London: Springer.

Good luck with your work!

Author Response

(The authors gave the same response as above.)

Round 2

Reviewer 2 Report

Dear authors,

Thank you for the revision of your paper. My main issues have been properly tackled and I am satisfied with the overall revision of the paper.

Nevertheless, I still have some comments related to minor issues I detected reading the manuscript.

References

The first (and more relevant) one is related to the references. It seems that some problems arose during the revision. For example:

First case:

Old version

Line 43: In recent years, we can see enormous attention being paid to upcycling activities [4,5].

4.Singh, J; Sung; K; Cooper, T; West, K and Mont, O. Challenges and opportunities for scaling up upcycling businesses – The case of textile and wood upcycling businesses in the UK, Resources, Conservation and Recycling, 2019. 150. https://doi.org/10.1016/j.resconrec.2019.104439.

5. Villa, B; Nogueira, M; Callegaro and Ghezzi. D. Innovative and sustainable business models in the fashion industry: Entrepreneurial drivers, opportunities, and challenges. Business Horizons, 2017, 60(6), pag. 759-770.

New version

In recent years, we can see enormous attention being paid to recycling and upcycling sustainable related activities [4].

4. Cagno, E.; Neri, A.; Trianni, A. Broadening to sustainability the perspective of industrial decision-makers on the energy efficiency measures adoption: some empirical evidence. Energy Efficiency, 2018, 11(5), 1193-1210. doi.org/10.1007/s12053-018-9621-0.

Second case:

Old version

Line 85: In recent years, literature on sustainability transitions in the specific field of  business organization and management has emerged that poses a multi-level perspective (MLP) to capture the dialectical relationship between micro-level actors and macro-level structures (landscapes, regimes and niches) [11].

11. Geels, F. Technological transitions as evolutionary reconfiguration processes: a multi-level perspective and a case study, Research Policy, 2002. 31, 1257- 1274.

New version

In recent years, literature on sustainability transitions in the specific field of business organization and management has emerged that poses a multi-level perspective (MLP) to capture the dialectical relationship between micro-level actors and macro-level structures (landscapes, regimes and niches) [15].

15. Smith, A and Raven, R. What is protective space? reconsidering niches in transitions tot sustainability, Research Policy, 2012, 41: 1025- 1036.

Typos

I detected a very minor typo, Page 4: “Geels [11] characterize these interactions”, as there is a missing “s”. The paper is very well written, my suggestion is to have a carefully final read of the manuscript and maybe additionally use a grammar check software (no need for professional proofreading).

Also see references 15 highlighted in the previous Table.

Layout

At pag. 10, the section within the 2 quotes is not aligned with the rest of the text (right side).

Table 2 should be tidied up, as there are missing “-“ as in South Africa, double “-“ as in Brazil, different spaces before and after the text

Author Response

We would like to thank the editor and reviewers for their comments on our paper and their very useful suggestions for minor revisions.

We have revised the paper to address fully these outstanding concerns. Specifically we have improved the references, typos and layout. These changes have been highlighted in red. 

If you require any further information, please do not hesitate to ask.

Yours sincerely

Sara Calvo